# Rethinking performance crises in professional soccer: German coaches' insights into systemic vulnerabilities and escalating dynamics

Constantin Rausch[1]*, Julian Fritsch[2], Stefan Altmann[3,4], Jan Spielmann[4], Lena Steindorf[4], Darko Jekauc[1]

**1** Department of Health Education and Sport Psychology, Institute of Sports and Sports Science, Karlsruhe Institute of Technology, Karlsruhe, Baden-Württemberg, Germany, **2** Section of Sports Psychology, Department of Sports Sciences, Goethe University, Frankfurt, Hesse, Germany, **3** Department of Social and Health Sciences, Institute of Sports and Sports Science, Karlsruhe Institute of Technology, Karlsruhe, Baden-Württemberg, Germany, **4** TSG ResearchLab gGmbH, Zuzenhausen, Baden-Württemberg, Germany

* constantin.rausch@kit.edu

## Abstract

Professional soccer, as a global phenomenon, is characterized not only by outstanding performances but also by frequent and sometimes prolonged periods of underperformance, which represent performance crises. Despite the growing body of research on performance crisis, the specific perspectives of coaches remain underexplored. Previous studies have largely focused on players' viewpoints, resulting in a lack of understanding of the systemic and escalating dynamics of crises from the perspective of those in leadership positions. This study addresses this gap by investigating how professional soccer coaches perceive the development and persistence of performance crises in professional soccer. Employing a qualitative research design, twelve professional coaches with diverse roles (head coaches, assistant coaches, and goalkeeper coaches), a mean age of 43 years (range: 32–51), and extensive coaching experience in professional soccer (7–23 years, $M = 15.5$) were interviewed. The interviews were analyzed using data-driven and concept-driven content analysis based on thematic qualitative text analysis. The analysis reveals that performance crises are not attributable to isolated incidents but rather arise from the interplay between pre-crisis vulnerabilities, their transition to crisis dynamics, acute triggers, and escalating dynamics at team, organizational, and external levels. Pre-crisis vulnerabilities include latent factors such as organizational incongruence, and fragile team cohesion, that increase susceptibility to crises. Notably, coaches emphasized that success temporarily masks these underlying tensions, which surface and intensify when performance declines. Acute triggers refer to specific disruptive events or negative results that catalyze instability and initiate the crisis process. Escalating dynamics describe the self-reinforcing processes whereby psychological, social, and structural problems perpetuate and deepen the crisis. The study advances the field

**Data availability statement:** The data set consists of semi-structured interview transcripts from a qualitative study on performance crises in professional soccer. The transcripts include personal and organizational information that could potentially allow for re-identification despite anonymization. Data cannot be shared publicly due to ethical and legal restrictions imposed by Karlsruhe Institute of Technology (KIT) per participant-approved data privacy agreements with the KIT Data Protection Office. Participants provided informed consent for the scientific analysis of their interviews but did not consent to the public sharing or archiving of full interview transcripts. Qualified researchers may request pseudonymized data via institutional procedures: dsb@kit.edu.

**Funding:** The author(s) declare that financial support was received for the research and/or publication of this article. We acknowledge the support by the KIT-Publication Fund of the Karlsruhe Institute of Technology. The funders had no role in study design, data collection and analysis, decision to publish, or preparation of the manuscript.

**Competing interests:** The authors have declared that no competing interests exist.

by highlighting systemic, self-reinforcing cycles of crisis and organizational incongruence that undermine coaching authority. These insights have practical implications for improving leadership coherence and resilience strategies in professional sports environments.

## Introduction

Soccer is one of the most popular sports in the world, representing a global phenomenon with cultural, social and economic significance. This broad appeal is exemplified in the 2022 FIFA World Cup final, which drew 1.5 billion viewers, becoming the most-watched single sports event in history [1].

This widespread fascination for soccer could arise from its unpredictable nature [2], exemplified by unexpected outcomes like Leicester City's 2015/16 Premier League victory. Conversely, some teams fall short of expectations for an extended period. For example, during the 14/15 Bundesliga season, Borussia Dortmund was at the bottom of the table by matchday 19, with eleven losses and four draws, despite finishing second in the previous season. If not effectively managed, such situations can affect the entire organization of a soccer club. Relegation, for example, can have far-reaching consequences, involving loss of revenue and reputation, and ultimately leading to existence-threatening debts, license withdrawals, and insolvencies [3,4].

The shifts in team performance underscore the complex interplay of psychological, organizational, and contextual factors that characterize what sport psychologists define as performance crises [5,6]. Historically, Bar-Eli and Tenenbaum [7] initially used the term crisis in their theory of individual psychological crisis in competitive sports. Drawing on the inverted-U relationship between arousal and performance [8], the theory posits that both under- and over-activation increase the likelihood of entering a crisis phase.

Recently, the focus has expanded from the individual athlete to the collective level [5,6]. In this context, Buenemann et al. defined in their narrative review a performance crisis in team sports as a "continuous underperformance across multiple games, accompanied by team members' threat states and the inability of a team to effectively cope with this threat, resulting in low team functioning" [5 p826]. Based on this definition, the authors developed a process model divided into four distinct phases: Stage 0 includes stable traits like collective efficacy and team cohesion that influence a team's susceptibility to performance crises. Stage 1 identifies triggers such as attributions, expectations, and perceived consequences. A negative combination of uncontrollable (internal and stable) attributions, unfulfilled high expectations, and the perception of severe and far-reaching consequences of failure, can significantly impair an individual's ability to manage psychological pressure. Stage 2 covers coping strategies, which can lead to either recovery or deterioration. If maladaptive patterns persist, the crisis will progress to Stage X, the downward spiral.

In a similar vein, Jekauc et al. [6] described a performance crisis as a downward spiral based on a qualitative study with nine professional soccer players. However, unlike Buenemann et al. [5], they identified negative affective states resulting from

unmet expectations as the initial trigger of a performance crisis. These negative affective states subsequently initiate a cascade of psychological processes at both the individual (e.g., reduced self-confidence, rumination) and team (e.g., conflicts, disturbed communication) level. These processes within the individual players and the team reinforce each other, impacting on-field behavior (e.g., adopting a defensive mindset) and reducing the likelihood of meeting one's expected performance. This, in turn, creates a self-reinforcing cycle, as the probability of continued defeats increases, perpetuating the cycle anew. In this model, the coach plays a central role because their behavior can either escalate or de-escalate the situation.

These models of performance crises can be enriched by broader theoretical perspectives that emphasize multi-level interactions and interpretive processes. From an ecological systems viewpoint [9] and key principles of complexity theory, such as interconnectedness, emergence, and non-linear feedback [10], crises emerge from dynamic interdependencies across nested environments. Here, micro-level triggers can cascade into meso-level systemic threats, as articulated by Szymanski and Weimar [4], who identify deviations from expected team performance as key drivers of insolvency risk in professional soccer. Complementarily, sensemaking theory [11] describes a retrospective process in which individuals interpret ambiguous cues. Weick states, that individuals act ahead of cognition, "acting their way into belated understanding" [12 p419]. Within this framework, in times of performance crisis, a consensus may not be reached by club officials, coaches, and players due to divergent realities being experienced, exacerbated by mismatched performance expectations. These divergent realities could lead to breakdowns, such as coach dismissals and self-reinforcing instability, hindering learning and resolution [13].

Benz and Gehring [14] stated that the reasons for the lack of sports success are insufficiently investigated, leading to a situation where valuable lessons are not learned from past crises. They emphasized that soccer clubs often appear to be trapped in a cycle of recurrent coach dismissals. This tendency can be explained through the scapegoat theory, which posits that coaches serve as symbolic scapegoats to displace frustrations from stakeholders, without sustainably improving performance [15]. Galdino et al. [16] qualitatively explains these decisions through the social construction of leadership, driven by excessive media pressure, dominant politics in club administration, lack of strategic planning, illogical expectations, and player disengagement. Empirically, Flores et al. [17] analyzed dismissals of Argentine soccer coaches under pressure and found no performance gains. Similarly, Heuer et al. [18] demonstrated null effects in the German Bundesliga due to regression to the mean. Together, these findings highlight the need for a more comprehensive understanding of performance crisis dynamics in professional soccer.

The findings of Jekauc et al. [6] provide initial insights into these dynamics and placed the coach as a central role in managing a performance crisis. Although coaches play a central role in managing performance crises [19], their perspectives have been largely overlooked in previous research. Given that performance crises have adverse consequences for all stakeholders involved, an examination from the player's perspective alone may provide a limited understanding of the phenomenon. Coaches, as central figures in decision-making, emotional regulation, and communication within a team, are uniquely positioned to observe and influence both the onset and progression of performance crises [19,20]. Coaches provide access to dimensions of the crisis experience that are inaccessible through players' perspectives alone [20,21]. Incorporating coaches' insights is therefore essential to develop a more nuanced and integrative understanding of performance crises. Accordingly, the objective of the current study was to explore coaches' perceptions regarding the development and maintenance of performance crises within the specific context of German male professional soccer.

## Materials and methods

We applied the Standards for Reporting Qualitative Research (SRQR) checklist to account for all aspects of qualitative research [22] (see S1 Table).

## Philosophical and methodological orientation

A qualitative research design was used following the ontological relativism and constructivist epistemology detailed in Rausch et al. [19]. For a comprehensive description of the research philosophy and methodological orientation, see Rausch et al. [19]. Accordingly, this study is framed as qualitative, exploratory research, following Stebbins's tradition [23]. This approach involves a broad, purposeful, and systematic method to maximize discovery. The goal is to gain a better understanding of an understudied social phenomenon. In this case, the discovery includes coaches' perceptions of systemic vulnerabilities, triggers, and escalating dynamics in performance crises.

Since the cyclical framework of Jekauc et al. [6] served as the theoretical foundation for this study, the data analysis followed an abductive approach, which integrates both inductive and deductive reasoning. Rooted in the pragmatist philosophy of Peirce [24], abduction balances openness to empirical data and theory, allowing iterative refinement of theories based on observations. This approach is especially useful when findings challenge existing assumptions (for an example see S2 File), supporting theory development and/or refinement through a dynamic, reflective process [25,26].

## Sampling and participants

Prior to data collection, ethical approval was granted by the Ethics Committee of the Karlsruhe Institute of Technology. Coaches were contacted via email or telephone and invited to volunteer for this study. A combination of convenience, purposive, and snowball sampling was employed to access a hard-to-reach population of professional soccer coaches with limited availability. The first two coaches were recruited based on availability (convenience sampling), while the remaining ten were selected using a combination of purposive and snowball sampling techniques. Snowball sampling was initiated at the end of each interview, where participants were asked whether they could refer to other coaches who might meet the study's inclusion criteria. If referrals were provided, these individuals were first screened to ensure they met the following criteria: (a) they must have experienced a performance crisis during their coaching career, and (b) they must be considered professional coaches, as these coaches' expertise and specialized knowledge enabled information-rich data [27]. To ensure that participants had indeed experienced a performance crisis, the lead author reviewed each coach's career history prior to the interview (e.g., media reports, match records) and explicitly inquired about crisis experiences during the interview. Coaches were classified as professional if they had coached a team within one of Germany's top three soccer divisions, coached a national team, or held the UEFA Pro License, each indicating a high level of professional qualification. While all other coaches clearly met the professional criteria by coaching in one of Germany's top three divisions, one coach required further consideration, as he coached a team in the fifth division. However, despite coaching in the fifth division, the coach was leading a historically successful club with high public expectations and media attention. He had navigated the team through a period of intense performance pressure and internal instability, providing him with relevant experience comparable to that of coaches in higher divisions. Importantly, the lead author did not have any personal connections with these coaches.

This procedure allowed for the identification of twelve relevant participants while maintaining the study's focus on professional-level performance crises. All participants were male coaches from German professional men's soccer leagues. Their ages ranged from 32 to 51 years ($M = 42.92$, $SD = 5.87$), and their coaching experience ranged from 7 to 23 years ($M = 15.5$, $SD = 5.2$). The sample comprised various coaching roles, including eight head coaches, two assistant coaches and two goalkeeper coaches. Additionally, the sample included coaches with various qualifications: nine UEFA Pro Licenses, one UEFA A License, one UEFA Goalkeeping A License, and one coach without an UEFA License. Recruitment was concluded after twelve interviews, which provided sufficient depth and richness of data to address the research questions.

## Data collection

The interviews were conducted from 11/28/2022 until 03/16/2023. The duration of the interviews ranged from 39 to 97 minutes ($M = 68.92$; $SD = 18.49$). The interviews were carried out in German and the quotes were translated into English

for the presentation of the results. At the beginning of each interview, participants were informed about the general purpose of the study. Coaches were assured that the data would be treated confidentially and anonymously. Furthermore, they were informed that participation was voluntary and that withdrawal was possible at any point during the interview. All participants had previously given their written consent to participate as well as agreed to the audiotaping of the interview. After this initial briefing, semi-structured interviews were conducted. This format was chosen because it offers a balance between comparability across participants and openness to new insights. Due to geographic distance, online interviews were conducted with eleven coaches via video call, while one interview took place in person.

**Interview guide.** The semi-structured interview guide employed in this study was developed based on the theoretical framework concerning performance crisis in professional soccer as outlined by Jekauc et al. [6]. Following the initial draft, the guide was revised by the research team to ensure comprehensiveness and feasibility [28]. To further refine the guide, a pilot test was conducted with one active coach and one active player, both recruited through the researchers' professional network. The coach interviewed for pilot test was not among the final twelve coaches analyzed. Although the main study was conducted exclusively with coaches, the inclusion of a player in the pilot phase was intended to ensure that the questions were comprehensible and relevant from the perspective of those directly affected by coaching during performance crises. This cross-perspective feedback helped improve the clarity and contextual sensitivity of the guide. Based on the feedback, several modifications were made, including the addition of new questions to address previously overlooked aspects and enhance the depth of inquiry.

The revised guide consisted of five sections: an opening question to build rapport and gather background information, followed by three sections of open-ended and follow-up questions exploring coaches' understanding, detailed experiences, and perspectives on performance crises. For example, in the third section, participants were asked to describe a specific performance crisis they had encountered in detail (e.g., "Could you describe the performance crisis you experienced in all its facets?"), followed by targeted questions addressing contributing factors and contextual influences (e.g., "What factors contributed to sustaining this crisis?"). The final section invited any additional topics from participants. The full guide can be found in the supporting information (S3 Table).

## Data analysis

All interviews were audio recorded and manually transcribed verbatim by the first author using the software f4 (dr. dresing & pehl GmbH, Marburg), resulting in 190 pages of single-spaced text. The data were analyzed using a data-driven and concept-driven content analysis based on thematic qualitative text analysis by Kuckartz [29], guided by the cyclical framework of performance crisis [6]. All coding was conducted using f4, which facilitated code management. A detailed description of the analysis process, based on the five-phase model proposed by Kuckartz [29], is provided in the supporting information (S4 File).

## Trustworthiness

Trustworthiness was ensured in accordance with the guidelines proposed by Elo et al. [30]. Particular attention was given to enhancing the credibility of each phase of the qualitative content analysis, including *preparation*, *organization*, and *reporting* of results.

In the *preparation* phase, several measures were taken to ensure the trustworthiness of the study. First, semi-structured interviews were selected as the preferred method of data collection, as the study pursued both data-driven and concept-driven aims. Second, the development of interview questions involved a "critical reference group" [31], meaning that the questions were co-constructed with individuals for whom the research is intended to be meaningful and beneficial. Given that performance crises affect multiple stakeholders, the interview guide was developed in collaboration with researchers specializing in sport psychology and practicing sport psychologists, and subsequently refined through feedback from one active player and coach. As Elo et al. emphasize, involving a critical reference group can help ensure that

researchers are "asking the right questions in the right way" [30 p4]. Finally, since coaches are central to the theoretical framework proposed by Jekauc et al. [6], purposive sampling was employed to recruit participants with the most relevant expertise. A diverse sample of coaches was selected to capture a range of perspectives on the phenomenon of performance crises, thereby enhancing the credibility and richness of the data.

Trustworthiness in the *organization* phase mainly focused on consensual coding [32]. In alignment with our ontological relativism and constructivist epistemology, intercoder reliability was achieved through consensus coding [32], which acknowledges the inseparability of the coding process from researchers' subjectivities [33]. Accordingly, CR and a research assistant engaged in frequent critical dialogues and weekly meetings to reflexively discuss, examine, and resolve coding discrepancies until a shared understanding was achieved. This collaborative process was further supported by DJ, who acted as a "critical friend" [33], challenging assumptions and encouraging reflexivity by questioning interpretations and prompting deeper analytical engagement. Furthermore, concept mapping [34] was conducted using freehand drawings on paper during in-person interviews and shared screens with drawing tools during online interviews. These maps were included in the coding dataset as visual presentations of the structure of various concepts and patterns, having been scanned or saved beforehand.

In the *reporting* phase, the study aimed to achieve resonance, defined as the capacity of research to meaningfully echo with its audience [35]. Rather than relying on formal generalizations, qualitative research seeks resonance through transferability [36] or naturalistic generalization [37], both of which are determined by the reader. Resonance is achieved when the research provides a vicarious experience, and the story of the research overlaps with the reader's own situation [35]. In this sense, the reader is encouraged to judge whether the results are transferable to other contexts and resonate with the reader.

## Findings

### General overview

Performance crises in professional soccer do not arise from isolated factors, but from the interplay of pre-crisis vulnerabilities, acute triggers and escalating dynamics at individual, team, organizational and external levels (Fig 1). Rather than unfolding in a linear sequence, these components interact continuously. Pre-crisis vulnerabilities increase susceptibility to disruption, acute triggers catalyze instability, and mutually reinforcing escalation processes perpetuate dysfunction. Coaches' accounts reveal that this network of interacting influences, shaped by dynamic relationships among psychological, social and structural dimensions, drives both the emergence and persistence of a crisis. The following sections will describe these three core components in detail, exploring how performance crises develop, escalate, and persist in professional soccer.

### Definition of performance crisis

Following the statements of the coaches, six main categories could be identified using a category-based analysis that ultimately led to a definition of performance crisis in soccer, which states:

> *A performance crisis in professional soccer is a prolonged state in which both results and performance fall significantly short of expectations, characterized by a loss of control on an individual level, dysfunctional team dynamics, and destabilizing external conditions.*

A performance crisis in professional soccer was described as "when you fall short of general expectations over a longer period" (Coach 11). It is not solely characterized by objective failures such as defeats or poor league standings, but rather emerges from the interplay of multiple factors at the individual, team, organizational, and external levels. At the individual level, a crisis often manifests as psychological strain, self-doubt, a sense of overload, or loss of control as expressed by Coach 4: "You don't have the solution at hand in the moment [of crisis]. In the worst case, you start trying lots of things

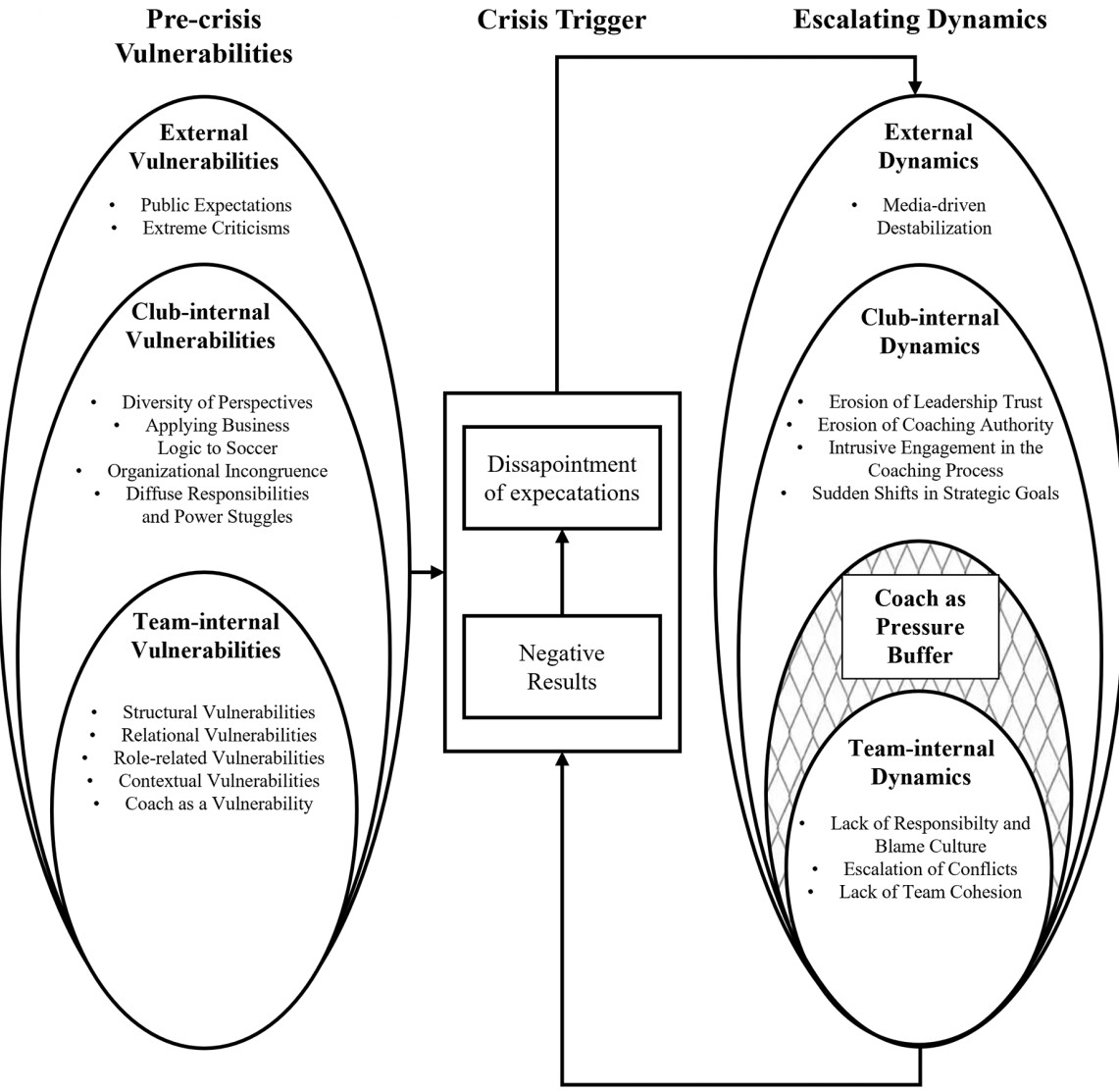

**Fig 1. General overview of the findings: Vulnerabilities, triggers, and escalating dynamics in professional soccer.**

and don't get anywhere, which reinforces that feeling of being helpless". At the team level, dysfunctional processes may arise, including disrupted communication, lack of collaboration, internal conflicts, or a decline in team cohesion: "In crisis situations, you always face the problem that your team is no longer a unit" (Coach 9). Furthermore, the crisis is intensified by external pressure, such as media scrutiny, fan expectations, or demands from club leadership, all of which shape and reinforce the perception of crisis as expressed by Coach 10: "And in soccer, it's usually the case that external factors start to carry more weight. That means people from outside begin to exert strong influence when things aren't working".

## Pre-crisis vulnerabilities

Pre-crisis vulnerabilities describe underlying vulnerabilities that increase the likelihood for a crisis to emerge: "[Pre-crisis vulnerabilities are] factors that lead to continued defeats". Players operate within a high-pressure environment in which

influences from external sources (e.g., media, fans), club-internal structures (e.g., leadership dynamics), and team-internal processes (e.g., cohesion, role clarity) converge. These vulnerabilities often operate subtly and may not immediately impair performance. Coach 5 described these vulnerabilities as "micro-crises", noting:

> For me, these would be what I'd call micro-crises, because they're very small factors that may not last very long but still create stress for a team. And in the worst case, they might even influence results, maybe just for one, two, or three matches, but I definitely believe they have an impact on the atmosphere within a club and its team.

However, when they converge or intensify, especially under pressure, they can significantly disrupt team dynamics and undermine the ability to respond effectively to challenges:

> But when those little things start piling up, like, more and more players begin to question stuff, some feel hurt, some lose trust. (…) That's when it starts to snowball. And if it gets too much, too intense, that's when things really break apart. (Coach 2)

Crucially, these vulnerabilities also reduce a team's probability of success in the short term, thereby increasing the likelihood of disappointing results. Therefore, these vulnerabilities have the potential to significantly influence the interpretation of both success and failure and play therefore a defining role in shaping expectations of upcoming matches. The following sections will go into more detail about pre-crisis vulnerabilities at the different levels.

**External vulnerabilities.** External influences such as media, fans, and personal networks were frequently described by coaches as pre-crisis vulnerabilities. While these influences originate outside the club and cannot be directly controlled, they significantly affect players' mindset, team dynamics, and overall focus: "And they [consultants, families, media, friends] influence the mindset and the mood of the players and of course that influences the team" (Coach 10). Frequently cited subcategories were *public expectations* and *extreme criticism*.

Public expectations play a crucial role in shaping the climate during performance crises. External parties can create unrealistic expectations that could lead to an imposed pressure to succeed. Coach 12 mentioned that because the team was in a relegation position, the expectation was created that they had to win every match to avoid relegation: "Everyone said it, the press said it, the fans said it: 'You have to win every match from now on'". This consequently can lead to performance pressure among the players, as they perceive that failing to meet expectations will lead to undesirable consequences.

In addition to these public expectations, extreme criticism can significantly impact team dynamics. The public exposure faced by players and coaches is a key aspect of this dynamic. Coach 7 emphasizes this public exposure:

> The crowd starts booing right after the match. Then you check social media: I do not really do that myself, but if you were to read the comments there, they insult you in the worst possible ways, you know. The press just tears you apart. And if you also have a sports director who adds pressure one could easily crumble under all that, if not resilient enough.

Such criticism is not only harsh but often occurs prematurely, as Coach 4 emphasized: "The press and the media, I definitely had the feeling that criticism was expressed very quickly and with a certain sharpness". Coach 6 suggests that this external perception rarely aligns with the internal reality of the team and the players: "People aren't there every day, after all. So, what you read and hear after the matches is often insane, because you have a completely different insight into the team".

**Club-internal vulnerabilities.** Club-internal vulnerabilities refer to structural and relational issues within the organization, such as misalignment, poor coordination across leadership levels, and unclear responsibilities. Coaches

described how the absence of coherent guidance and shared purpose can leave the organization directionless. As Coach 12 explained: "If you don't have a lighthouse, if you don't know where you're going, then the ship just drifts. (…) If you don't have a clear direction, a reason why you're doing all this, then you have a problem". This sense of disorientation illustrates how weak strategic alignment and inconsistent leadership communication can undermine stability and decision-making. Commonly identified club-internal vulnerabilities include *diversity of perspectives*, *applying business logic to soccer*, *organizational incongruence*, and *diffuse responsibilities and power struggles.*

A central theme reported by coaches was a diversity of perspectives among key stakeholders. As one coach described, "accordingly, the perspective [referring to the perspectives of the board, the sports director, and the coach] was simply different (Coach 4)". Such differences were generally seen as an inherent, as Coach 2 pointed out: "And even when things are going well, there are still small or isolated cases where there may be problems. Where the relationship suffers, where there may not be 100% trust. That always happens". However, while these differing perspectives exist prior to performance crises and are not necessarily problematic, this diversity may amplify tensions, as misaligned expectations and interpretations hinder coordinated action and mutual trust. For example, Coach 4 described how these different perspectives often led to contrasting perceptions and interpretations of team performance and were not only rooted in expertise, but also in proximity to the team: "Sometimes those expectations were contradictory. The sports director, who watched every training session, knew what the team was capable of. But the president saw something completely different and spun his own narrative".

Common explanations for these tensions were different backgrounds, professional logics, and expectations among stakeholders. The diversity of perspectives becomes particularly visible when individuals from business-oriented environments apply their own mindset to the world of professional soccer. Applying business logic into the world of professional soccer can quickly lead to frustration or increased pressure, because success in professional soccer is less predictable than in other areas:

> There are many factors that influence success in soccer, like experience, injuries, training processes, and internal team dynamics. These can only be planned to a limited extent, even by coaches or clubs. That's why success in soccer is less predictable. But since soccer clubs are often led by people from the business world, they're used to thinking success is planned and then achieved. So, their expectation is: if we invest money, there must be success. But that direct link doesn't exist in soccer the way it does in business. (Coach 9)

Coaches perceive that applying business logic to soccer often leads to a narrowing of interpretation, where performance is reduced solely to the outcome of the match. These diverging perspectives not only shape how performance is interpreted, but also influence key decisions within the organization, often in ways that reflect limited domain-specific expertise. As Coach 11 explains:

> Boards and presidiums often aren't soccer experts. But they still make decisions about the coach's quality. It's strange, these are successful people in their fields, but not necessarily in soccer. Yet they decide whether a coach stays or goes. In professional soccer, a coach's quality is judged solely by points. (Coach 11)

These divergent perceptions not only shape how situations are interpreted, but also reveal deeper structural issues within the organization, summarized as organizational incongruence. When there is no shared understanding of the situation, coordination is fragmented. Organizational incongruence is a visible indicator that perspectives within the club are not coherent: "We [sports director and head coach] were never on the same page. We never acted in sync." (Coach 7). These communication gaps were highlighted by a lack of trust and coherence across hierarchical levels. Coach 12 puts it: "There were discrepancies between the different levels. Definitely no unified language. No trust between us" (Coach 12).

One central theme emphasized by coaches was diffuse responsibilities and power struggles within the club's leadership structures. Coach 3 described this diffuse structure as follows: "It was a club structure where far too many people were giving directives who had nothing to do with the sporting side." In this context, unclear hierarchies and competing interests led to instability: "There were many people at the top who wanted to have a say, and political power struggles were being played out. Ultimately, this came at the expense of the club and its performance" (Coach 5).

**Team-internal vulnerabilities.** Team-internal vulnerabilities refer to structural, relational, role- and coach-related as well as contextual dynamics within the team that may undermine stability and collective performance. These vulnerabilities represent latent weaknesses that can exacerbate the impact of external stressors or trigger internal instability when the team encounters difficulties. The coaches noted that problems from the club or outside environment often surface within the team and can quickly turn into performance crises: "I do believe that this turmoil [at the club level] was the reason why so many players within the team started heading in completely different directions" (Coach 3). Commonly identified team-internal vulnerabilities include *structural*, *relational*, *role-related*, and *contextual vulnerabilities* as well as the *coach as a vulnerability*.

Structural vulnerabilities are factors that weaken the team at a structural level and are difficult to change. Commonly reported examples include an unbalanced team composition, a lack of experience and/or resilience, chronic overload, and susceptibility to injury. These factors may not immediately lead to performance breakdowns, but they can create a fragile foundation that becomes critical when additional stressors arise:

> That meant that we were completely exhausted at the end of the first half of the season, and it all started back in November. So, the guys were just totally tired, completely drained, exhausted. And we could no longer maintain the intensity that used to characterize us. And as a result, we also could not bring in the results. (Coach 11)

Relational vulnerabilities refer to interpersonal tensions and dysfunctional patterns of interaction within the team. These vulnerabilities are often latent during stable periods but become disruptive when the team enters a crisis. When these relational tensions surface, they can erode trust, disrupt communication, and weaken the team's emotional resilience. Frequently cited examples include weak cohesion, underlying conflicts, and problematic relationship patterns within the team. As Coach 10 explains:

> In a squad of 20 to 25 players, not everyone always gets along perfectly. There are always differences, some players just don't connect on the same level. So naturally, people interact with each other in different ways. But when things are going well, that's usually not an issue. It only becomes a problem when you hit a crisis. Then suddenly, those differences come to the surface. (Coach 10)

Role-related vulnerabilities describe uncertainties and ambiguities regarding players' roles, responsibilities, and tactical expectations within the team. These vulnerabilities emerge when communication is inconsistent, roles are poorly defined, or players lack a clear understanding of their place in the team structure. Such conditions may not immediately impair performance but can gradually destabilize team functioning and contribute to the onset of a crisis:

> One major challenge we faced was that the players had a completely unbalanced understanding of their roles. They hadn't yet found their place in the team, and they didn't really know, or couldn't yet know, what their role was supposed to be. (Coach 4)

In addition, several uncontrollable contextual vulnerabilities were also mentioned by the coaches, such as bad luck, unfavorable referee decisions, and congested match schedules: "We simply had poor results. I'd even say we were a bit unlucky, with red cards and all the things that come with it" (Coach 10). While these are not inherently part of the team's

internal dynamics, they can act as contextual stressors that interact with existing team-internal vulnerabilities and potentially accelerate the development of a crisis.

Coaches acknowledged their involvement in crisis development through statements indicating varying degrees of responsibility. For example, one coach stated, "Yes, definitely. You have a big share in it. (…) However, your hands are also tied because you depend on sports management, you are also an employee" (Coach 7), while another said, "Sometimes it's 20%, sometimes 50%, sometimes maybe 90%. But he definitely has shares in it" (Coach 6). These accounts highlight the perceived significance of the coach's role as both a contributor to and subject of team crises.

### From pre-crisis vulnerabilities to crisis dynamics

Success often acts as a temporary stabilizer that masks pre-crisis vulnerabilities. According to Coach 7: "When you win, there's no crisis, everyone just bites the bullet", suggesting that winning can suppress emerging conflicts or dissatisfaction. When performance declines, previously hidden tensions within the organization can quickly surface. Coaches described how the absence of success acts as a catalyst for criticism and conflict across all levels: "When the results aren't there, people around you suddenly find plenty to criticize" (Coach 10). Similarly, Coach 7 emphasized the accumulating pressure beneath the surface: "When there is no sporting success, even if it doesn't show on the outside, there is always potential for conflict (...) That's why the potential for conflict is extremely high when you lose." These reflections illustrate that while success may temporarily mask existing differences, it does not resolve them. A central reason why these tensions intensify in times of declining performance lies in the gap between expectations and actual results. This mismatch between expected and actual outcomes not only fuels conflict but also marks a critical turning point in the development of a performance crisis.

### Crisis trigger

Crisis triggers refer to acute events or developments that catalyze the development of a performance crisis: "There's a trigger, and that sets off a whole chain of other things" (Coach 10). These triggers often take the form of negative match outcomes, unmet performance expectations, or critical incidents that expose and activate underlying tensions. From the coaches' perspective, such events are rarely isolated but acquire significance through accumulated pressure from various stakeholders. As Coach 7 noted: „Yes, that was on a certain match day when we lost. It was in [name of the club]. And then, wow, so many things happened, so many incidents. We lost the game, and at that moment, I knew we were in a crisis". In this sense, crisis triggers function as psychological and organizational turning points: they intensify scrutiny, shift internal dynamics, and mark the moment at which pre-crisis vulnerabilities manifest into visible instability.

**Negative results and expectations.** Negative results were consistently identified by coaches as a key turning point that marks the transition from pre-crisis vulnerabilities to a self-reinforcing dynamic, in which tensions intensify, and the crisis begins to escalate. As Coach 6 noted "especially in professional soccer, everything depends on whether things go well or poorly, but in the end, it all comes down to the weekend's result". This is also illustrated by the statement of Coach 1: "If you start losing matches, that puts you in a crisis".

The reason for this is that persistent match losses manifest themselves in the threat of goal achievement and the failure to fulfill expectations. As Coach 11 explained, "you simply fall short of expectations over a longer period of time," adding that "the higher the expectations, the faster the fall", reflecting that high expectations can accelerate the perception of a crisis. Still, not every defeat automatically triggers a performance crisis.

**Differentiation between result crisis and performance crisis.** Several coaches emphasized the importance to differentiate between a result crisis and a performance crisis. A result crisis refers to a state of consecutive negative match outcomes, in which the overall performance and the team dynamics remain intact: "A result crisis is when the team's performance, despite the lack of positive outcomes, still aligns with the coach's expectations in terms of tactical execution,

quality, and potential" (Coach 8). In such cases, the result crisis is primarily outcome-driven and may be attributed by the coaches to external factors such as opponent strength, randomness, or missed opportunities.

A performance crisis, in contrast, is characterized by prolonged negative results due to a performance drop and internal dysfunctions. Here, Coach 7 explains that after a few defeats, the focus shifts from external attributions to the internal functioning of the team:

> When you lose two games, tactics no longer matter. What you start to see is the inner life of the team. You can tell by the way they perform. If they're still playing well despite the losses, that's fine. But if the performance drops too, then you can tell, something's not right within the team.

Based on this statement, team performance can be understood as a mirror of the team's internal dynamics. Thus, declining performance may not only reflect technical or physical deficits, but also deeper psychological and relational processes within four different levels, which will be described in the following.

### Escalating dynamics

While pre-crisis vulnerabilities represent latent weaknesses that increase the susceptibility to crisis, escalating dynamics capture the processes of intensification and maintenance. Once initial stressors or performance drops activate these underlying vulnerabilities, they can set in motion self-reinforcing processes across different levels (e.g., external, club-internal and team-internal). The escalating nature of these dynamics is reflected in Coach 10's description, showing how initial negative results can trigger a chain of interpersonal and organizational disruptions: "At first, only the match outcomes were negative, but later, disagreements emerged between the people, leading to mood issues, discrepancies in content, and differences of opinion". These dynamics illustrate how crisis development is rarely linear: interactions between external influences, club structures, team processes occur in overlapping waves that amplify instability and complicate recovery. The following section provides a detailed overview of the three levels at which these escalating dynamics unfold.

**External dynamics.** External dynamics refer to influences outside the team structure that can contribute to the intensification of a performance crisis. These factors often create additional layers of pressure and distraction, making it harder for teams to maintain focus and performance. Coach 10 illustrates this by describing how ongoing external distractions during a difficult phase made it hard for the team to concentrate and work effectively: "We couldn't really concentrate fully on our work. There were always side issues that dragged on from January to July. We were never able to work in peace the way I would have liked". In particular, *media-driven destabilization* was mentioned by coaches as an external force that can intensify internal uncertainty and accelerate the crisis process.

Media-driven destabilization describes how media reports can undermine trust within the team and create uncertainty for the coaches. Media sources create narrative conflicts that strain the team structure, even unintentionally. As Coach 11 described:

> And then they keep trying to find conflicting statements. You can see it from weekend to weekend: they interview a coach, they interview a player, take two statements and try to play them off against each other. 'The coach said this and that, called it a stupid defeat, and the player says he's deeply dissatisfied – that's unacceptable,' and suddenly they try to create conflict where there is none.

Furthermore, some media companies are perceived as complicit in spreading pressure-inducing rumors, sometimes in collaboration with club officials. Coach 7 emphasized the influence of these alliances:

> [Name of the media company] has an incredible amount of power. Unbelievable, you can't even imagine. And that's why they all work together. They have to. So, [name of the media company] would never say, for example, that a coach is in crisis or on the brink unless they got that information from the club.

**Club-internal dynamics.**  Club-internal dynamics refer to structural, relational, and communicative dynamics within the organization that can contribute to the maintenance of a performance crisis. Coaches frequently described how pressure is transmitted from higher hierarchies to lower ones: "Then, the pressure goes from the top down. The management board passes the pressure on to the sports director, who then passes it on to the coach. The coach may pass it on to the team" (Coach 9). This top-down influence is also depicted by Coach 10, which illustrates how a lack of alignment and stability at the club level can erode clarity, trust, and direction at the team level, ultimately impairing psychological safety:

> Well, if the sports director holds a different opinion than the coach and makes that known, then of course the players start to doubt. They're no longer convinced of the path being taken. And if there's no conviction, if there's doubt, then we're back to things like psychological safety, self-confidence, and so on. It becomes a kind of ping-pong game between all parties.

This breakdown in communication is not merely interpersonal, but it reflects broader structural tensions within the organization. The following subcategories: *Erosion of leadership trust*, *erosion of coaching authority*, and *intrusive engagement in the coaching process*, illustrate how disrupted communication contribute to instability, undermine leadership roles, and fragment strategic alignment.

The erosion of leadership trust illustrates the consequences within the organization when divergent perspectives collide, particularly during phases of poor performance. During a performance crisis, communication between leadership levels can become fragmented, reactive, and emotionally charged due to overall organizational instability. These periods often reveal deep-seated issues in leadership communication, with trust emerging as a central theme. Coaches described how trust and collaboration with club leadership could deteriorate rapidly when performance declined: "You start losing games, and suddenly you're in a crisis, a crisis of trust" (Coach 1).

Here, trust was not a stable foundation but a conditional resource and withdrawn when short-term results failed to meet expectations. What had previously been a stable and respectful working relationship could shift overnight into scepticism and withdrawal of support as described by Coach 10:

> The relationship with part of the leadership [referring to the board and the sports manager] had been very good. But when the results crisis hit, the trust and high regard disappeared overnight. Everything was suddenly questioned. What had been seen as strengths were now seen as weaknesses. The collaboration changed completely.

The erosion of coaching authority often occurs when trust is withdrawn at the sign of short-term failure, and primarily describes the structural and systematic restriction of the coach's decision-making freedom and leadership role.. In this sense, coaches were not only "not supported," but also actively restricted in their decision-making freedom:

> At the time, I wanted to remove a long-serving player, the captain, from the team and I was forbidden from doing so. To put it bluntly: I couldn't make decisions in my own area, where I was supposedly the boss. (Coach 3)

Closely related, but conceptually distinct, is intrusive engagement in the coaching process, which focuses on the active and often excessive intervention from club leadership in the coach's day-to-day work. For example, one coach illustrates this engagement by describing how the sports director shifted from passive observer to active participant once the season started poorly. While this may reflect a desire to intervene, it could disrupt established leadership boundaries and underscores the absence of predefined roles. While such engagement may stem from a genuine desire to help, it can unintentionally signal a lack of trust, influencing the entire team as described by Coach 10:

> But that wasn't in the interest of the sports manager, who wanted to keep the player on the pitch. And of course, the team notices and feels that. As a coach, you're then forced to explain yourself and can't implement many of the things

the sports manager wants, because you first must deal with that issue. And that doesn't help your credibility, of course. It doesn't help team cohesion, and so on and so forth. These are additional internal problems that the whole situation brings with it.

Intrusive engagement in the coaching process often goes hand in hand with a broader inconsistency in organizational decision-making. This inconsistency becomes particularly apparent in sudden shifts of strategic goals during performance crises, where objectives are redefined without sufficient coordination or explanation. Coach 12 described his dismissal despite his team's phenomenal start to the new season. However, after a period of bad results, he experienced a sudden shift in expectations and support from the club's leadership:

Then the old mechanism kicks in. When the board suddenly has different ideas and completely changes the objectives. The goal was a top-half finish, maybe even top 6. That was the declared seasonal goal (…) Everything was thrown overboard because of emotion. The structure we had agreed on was suddenly forgotten.

**Team-internal dynamics.** During a performance crisis, team-internal dynamics often shift in dysfunctional ways, leading to emotional tension, fragmentation, and a breakdown of collective accountability. Coaches reported that the absence of positive results directly affects the team atmosphere: "The lack of results leads, more or less, to a change in the atmosphere" (Coach 6). This shift in team atmosphere, which was highlighted as the "most important thing" (Coach 9), could lead to various dysfunctional processes such as *lack of responsibility and blame culture*, *escalation of conflicts*, and *lack of team cohesion*.

Several coaches described situations in which their team lacked a sense of responsibility, leading to a breakdown in overall functioning:

What we found there was a team where no one could take responsibility. The whole thing had just collapsed. They were totally done. You could feel it right away. (…) No one stepped up during training, no one took charge. There was no organization, no self-management in so many areas.

This absence of responsibility did not just result in disorganization, but also fostered a culture of blaming others. Instead of individuals assuming responsibility, mistakes and failures were projected onto teammates: "There was constant finger-pointing. (…) It was always 'I'm doing my best, if it's not working, it's the others' fault'" (Coach 12).

Conflicts emerged as a central marker of crisis within teams. These conflicts, however, rarely remain on a purely professional level. As pressure mounts, discussions and confrontations tend to escalate and become personal: "The problem with discussions and confrontations is that at some point, it becomes personal" (Coach 7). Coach 7 illustrated this dynamic with a concrete example, describing how mistakes from one match are carried over into subsequent situations:

And if you lose next week, we have this one party, for example, where someone made a huge mistake on the next match day. Yes, then it [the conflict] comes up again. Then emotions come into play, then you yell at him, and the one you yelled at thinks, "Ah, he was just waiting for me to make a mistake anyway," which is why it's always difficult to turn things around.

A recurring theme in the coaches' accounts was the fragmentation of the team dynamic in moments of crisis. Coach 7 described how internal divisions can escalate to the point of active opposition: "At some point, the team basically split into three groups. Two of them were working against each other, blaming one another (…). The conflict had been going on for a long time, and it had become personal, really personal". Instead of pulling together to confront challenges, players drift apart, which prevents the emergence of shared resilience.

**Coach as pressure buffer.** Several coaches in this study describe that the head coach "is the face of the team" (Coach 6). This visibility makes the coach the primary target of external expectations from fans, media, and club management:

> The expectations were projected onto you as the coach, and you had to manage all the structures. The pressure from above, from the leadership, the pressure the team was under, maybe even internal problems that came up within the team, and so on. (Coach 10)

Several coaches emphasized the importance of shielding the team from this pressure and maintaining internal stability. Coach 1 highlights how such pressure is often passed on to the team without being filtered: "And this pressure is then passed on to the team without any filtering. For me, filtering that pressure would be the responsibility of a good leader". Coach 11 elaborates on this responsibility, noting that the coach must absorb and contain the pressure rather than transmit it to the players: "The coach has to try to absorb the pressure that exists (…) That's part of the job: to buffer that pressure".

Taken together, these perspectives illustrate the coach's role as a mediator between external expectations and internal team dynamics, acting as a stress buffer to protect the team's psychological resilience and performance capacity during periods of a performance crisis.

## Discussion

The objective of the study was to explore coaches' perceptions regarding the development and maintenance of performance crises. The findings reveal that performance crises in professional soccer are shaped by interdependent processes across individual, team, and organizational levels, which interact in non-linear and mutually reinforcing ways. Consistent with complexity theory's principles of emergence and interconnectedness [10], coaches described how minor vulnerabilities accumulate and interact to create systemic instability. Unlike the cyclical progression described in Jekauc et al. [6], coaches highlight that crises develop through the interplay of pre-crisis vulnerabilities, acute crisis triggers, and escalating dynamics across three levels: external, club-internal, and team-internal. This refinement enriches the understanding of performance crises by capturing the multi-level complexity of factors that predispose teams to crises and the mechanisms that drive their escalation.

### Pre-crisis vulnerabilities

Drawing on ecological systems theory by Bronfenbrenner [9], the identified pre-crisis vulnerabilities operate across multiple levels that shape team functioning. Teams are inevitably embedded in a complex system in which individual-level variables are nested within team contexts, which in turn are shaped by broader organizational and external environments [38]. Coaches in the current study illuminate how these systemic pre-crisis vulnerabilities could set the stage for destabilization before any trigger unfolds the crisis. This perspective of vulnerabilities being prevalent before the development of a performance crisis builds upon and nuances earlier phase-based models such as the collective team collapse framework [21], the multidimensional momentum model [39], and the performance crisis model in team sports [5], all of which highlight factors that increase susceptibility to negative outcomes.

In line with the terminology by ecological systems theory [9], external vulnerabilities such as public expectations represent exosystem influences that indirectly affect team dynamics, while club-internal factors like organizational incongruence reflect mesosystem disruptions. Team-internal vulnerabilities operate at the microsystem level, involving direct interpersonal interactions among players, coaches, and staff. This ecological framework demonstrates how performance crises emerge not from isolated incidents but from the dynamic interplay between nested environmental systems that either support or undermine optimal team functioning [40]. For example, public expectations and extreme criticism can

accelerate the crisis development process by injecting external narratives that increase uncertainty and pressure [41,42]. Furthermore, club-internal vulnerabilities, such as organizational incongruence and power struggles can diminish the team's strategic clarity and psychological safety [43]. Consequently, these pre-crisis vulnerabilities can create a fragile environment that increases team's susceptibility to destabilization, which could increase psychological strain among members, and ultimately impairs on-field performance [41,44].

Our findings identify coaches as a significant pre-crisis vulnerability, explicitly addressing an aspect not covered in the framework by Jekauc et al. [6]. Unlike their model, our findings highlight the critical function of pre-crisis vulnerabilities across external-, club-internal, and team-internal levels and underscores how coaches' ambivalence and difficulty in acknowledging their influence can contribute to team susceptibility to crises. Notably, this dynamic often results in problem-solving being delegated to the same actors (e.g., coaches) who contributed to the vulnerabilities, perpetuating ineffective interventions and vicious cycles that impose psychological strain on fans alongside tangible/intangible losses for stakeholders like employees, sponsors, and partners. This expands the understanding of crisis development by incorporating latent factors that set the stage for acute triggers, thereby refining and enriching the cyclical framework of performance crises in professional soccer.

## Crisis trigger

Crisis triggers often act through the underlying mechanism of expectations, which shape how acute events are perceived and interpreted within the team and organization. According to self-regulation theory [45], expectations serve as a reference point that determines whether negative outcomes are viewed as simply poor results (e.g., a results crisis) or as a performance crisis reflecting deeper issues within the team. As coaches described, when expectations are high or unrealistic, even minor failures can significantly intensify frustration and scrutiny, accelerating the transition from pre-crisis vulnerabilities to a crisis. This finding is consistent with the study of Szymanski and Weimar [4], who found that deviations from the expected performance are the crucial driver for insolvency in professional soccer. In this sense, the interplay between expectations and pre-crisis vulnerabilities creates a complex dynamic. Pre-crisis vulnerabilities such as extreme criticism, organizational incongruence, or role-related ambiguities can increase susceptibility to disruption, while unmet or unrealistic expectations can amplify the psychological consequences of triggers. This amplification could exacerbate emotional responses like frustration and anxiety, fueling adverse group dynamics and accelerating crisis progression [6]. Thus, expectations seem to act as a critical psychological mechanism that links acute triggers with pre-crisis vulnerabilities, shaping the trajectory and intensity of performance crises.

## Escalating dynamics

While Jekauc et al. [6] provided detailed insights into team-internal processes during performance crises, the coaching perspective in this study enriches the existing framework by providing a deeper understanding of how organizational factors interact with psychological and team-level processes, and by highlighting the critical role of organizational dynamics. This aligns with Jones [46], who argued in his reflections on excellence in sport and business, organizational issues may exert the greatest impact on performance. The club-internal dynamics described by coaches can be explained by the lens of sensemaking theory [11]. For example, organizational incongruence, diffuse responsibilities, or erosion of leadership trust identified in this study represent specific resilience deficits that amplify rather than buffer against performance-related stressors. These club-internal factors reflect a collapse of collective meaning-making processes [11]. When organizations fail to develop shared interpretations of critical situations, the resulting uncertainty can create fertile ground for crisis escalation. For example, coaches described how leadership behaviors, such as top-down transmission of pressure or the intrusive engagement in the coaching process, can reinforce the conditions it seeks to resolve. In this sense, organizational actors actively shape the performance crisis through their responses [47], creating fertile ground for systemic patterns of reactive decision-making.

From an ecological standpoint, this reactive decision-making is comprehensible, as scapegoating the coach is economically less costly and strategically more secure than confronting the squad. Players are reflected as intangible assets on balance sheets [48], drive broadcasting revenues through media value [49], and significantly enhance enterprise value [50], while coaches of higher-budget teams face elevated dismissal risks due to inefficient cost-per-point performance, reflecting their expendability relative to squad investments [51]. Especially "when the narrative demands a solution and the media chase in herds; when coaches become victim of the 'media-hunt'" [52 p169], scapegoating avoids terminating high-value player contracts, potential legal disputes, and controversy with fans/media emotionally attached to players over staff. This aligns with organizational theory where scapegoating thrives amid attributional ambiguity, enabling blame-shifting to individuals while avoiding costly squad restructuring [53]. These reactive interventions, exemplify the systemic pattern of 'fixes that fail' [54]. Coaches reported experiencing rapid leadership changes, such as the erosion of coaching authority and intrusive engagement in the coaching process. These changes were an attempt to provide temporary relief by helping the coach but created deeper structural problems. These findings provide empirical support for 'fixes that fail' that short-term solutions often neglect underlying causes, thereby perpetuating the dynamic club leaderships intend to solve [55].

As mentioned by Jekauc et al. [6], coaches play a key role in managing pressure and mediating team psychological processes during crises. This view is also supported by coaches, who consider them to be important pressure buffers. Coaches see themselves as mediators between external expectations and internal team dynamics, absorbing and regulating pressures to protect team cohesion and performance capacity. This dual function of the coach illustrates the complex and dynamic influence coaches exert during escalating crisis phases, aligning with the framework of Jekauc et al. [6].

## Implications

Building on organizational stress research in sport [56,57], enhancing organizational resilience should be a strategic priority for professional soccer clubs. Coaches described how club-internal processes, such as organizational incongruence, and institutional mistrust contributed to crisis escalation. These findings reflect key resilient organizational features described by Fasey et al. [43]: structural clarity, flexible improvement, shared understanding, reciprocal commitment, and operational awareness. Thus, systematically identifying and diagnosing pre-crisis vulnerabilities at all levels is a crucial first step. In the interconnected sports ecosystem, crises affecting professional soccer clubs can strongly impact surrounding stakeholders, including fans, employees, and sponsors, thereby amplifying their consequences. This interdependence highlights the importance of shared responsibility in strengthening collective resilience and preventing cascading failures [4]. Organizations could adopt tripartite stress management approach as described by Fletcher and Arnold [44]: primary interventions to optimize the organizational environment, secondary interventions to support individual stress responses, and tertiary interventions to minimize stress consequences. Implementing ecological, system-wide interventions [58] targeting multiple organizational layers simultaneously could strengthen resilience through cultural change rather than individualized coping strategies.

Communication mediates the positive effect on athletic performance and serves as a key regulator of team performance within teamwork processes, thereby predicting team success in complex environments like professional sports [38,59]. However, divergent perspectives during performance crises can lead to organizational incongruence and institutional mistrust. Since crises are socially constructed through interpretive interactions [60], all parties should share similar situational understanding for effective resolution. Fostering constructive communication channels and cultivating mutual trust emerge as key levers for mitigating crisis escalation and enhancing multi-level resilience.

A key insight from this study is that performance crises are often triggered by unmet expectations rather than poor performance alone. Coaches emphasized that the gap between expected and actual outcomes initiates the escalating dynamics, highlighting the need for proactive expectation management. In this sense, goal setting is closely linked to managing expectations, as goals serve as reference points shaping athletes' motivation and self-regulation [61]. Healy et

al. [62] emphasize collaboratively setting realistic, specific goals aligned with athletes' capabilities, implying the need for shared understanding between athletes and coaches. Extending this, this study suggests that clubs should systematically reflect on and align expectations across leadership, coaching staff, and players to ensure goals are realistic, transparent, and grounded in accurate assessments, thereby reducing pressure from mismatched expectations.

## Strengths and limitations

This study offers several methodological and conceptual strengths. First, it provides an innovative perspective by examining performance crises through the lens of professional coaches, who possess unique insights into the multi-level dynamics affecting team performance. Second, the sample of experienced professional soccer coaches from various competitive levels provided rich, contextually grounded insights that enhance the credibility and depth of findings. Third, the bounded focus on male German professional soccer coaches intentionally illuminates context-specific norms and structures, prioritizing qualitative transferability over statistical generalizability to other cultures [37]. Fourth, the structured qualitative approach, following a data-driven and concept-driven content analysis based on thematic qualitative text analysis, ensured comprehensive exploration of complex crisis phenomena.

However, there are limitations that should be considered. First, each coach has a different perception and perspective on a performance crisis. The proposed framework integrates diverse perceptions to reach common consensus but should be understood as one possible perspective shaped by specific contexts and subjective experiences rather than a definitive representation. Second, because performance crises are a highly sensitive topic that affects coaches, it is possible that despite rapport-building, coaches might not have been fully transparent in their responses.

## Future research

Based on this study, several directions for future research could be of interest. Firstly, this study should be seen as complementary to that of Jekauc et al. [6]. Complementing the research of the crisis from the players' perspective, new insights were obtained from the coaches' perspective. However, to draw a more comprehensive picture of a performance crisis, it would be fruitful to incorporate the perspectives of sport psychologists, fans, or club officials. Second, this study generates numerous testable hypotheses that warrant quantitative investigation through targeted research designs. Fruitful areas for investigation could include a longitudinal study tracking of professional soccer teams to identify predictive risk factors; an intervention study evaluating the effect of organizational resilience training on crisis frequency and recovery; and a multi-level analysis testing the relationships between expectation-performance discrepancies, escalating dynamics and crisis persistence across different stakeholders. Such designs could reveal mechanisms of sensemaking and escalation in real-time, advancing theory and informing interventions. Lastly, for a deeper understanding of the performance crisis and how it differs in various contexts, it would be useful to conduct similar studies in other types of sports as well as in women's professional soccer and other national soccer cultures, where differing norms around leadership, fan expectations, and organizational structures may shape distinct crisis dynamics [63].

## Conclusion

This study explored how professional soccer coaches perceive the development and maintenance of performance crises. Coaches view crises as dynamic, multifaceted processes shaped by the interaction of organizational, interpersonal, and psychological factors. The developed framework highlights pre-crisis vulnerabilities, and a self-reinforcing cycle of escalating dysfunction triggered by adverse results and unmet expectations. In this study, crises emerge through complex social interactions and organizational contexts, with coaches playing a key mediating role as pressure buffers. The findings emphasize the systemic nature of performance crises and the importance of enacted sensemaking in maintaining instability. The findings offer practical implications to guide organizational development, leadership alignment, and resilience-building efforts in professional sport environments.

## Supporting information

**S1 Table. Checklist for the Standards for Reporting Qualitative Research (SRQR).**
(DOCX)

**S2 File. An example of abductive reasoning in practice.**
(DOCX)

**S3 Table. Interview guide.** Note. Translation from German.
(DOCX)

**S4 File. Detailed description of the data analysis process.**
(DOCX)

## Acknowledgments

We would like to thank Nadine Engelmann, an undergraduate student who put a lot of effort into the project by helping with the coding frame, coding, and weekly discussions.

## Author contributions

**Conceptualization:** Constantin Rausch, Julian Fritsch, Stefan Altmann, Jan Spielmann, Lena Steindorf, Darko Jekauc.

**Data curation:** Constantin Rausch.

**Formal analysis:** Constantin Rausch, Julian Fritsch, Darko Jekauc.

**Investigation:** Constantin Rausch.

**Methodology:** Constantin Rausch, Darko Jekauc.

**Project administration:** Constantin Rausch.

**Software:** Constantin Rausch.

**Supervision:** Darko Jekauc.

**Visualization:** Constantin Rausch.

**Writing – original draft:** Constantin Rausch.

**Writing – review & editing:** Julian Fritsch, Stefan Altmann, Jan Spielmann, Lena Steindorf, Darko Jekauc.

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
