## [Decision Letter · Decision Letter 0]

6 Jan 2026

Dear Dr. Rausch,

Thank you for submitting your manuscript to PLOS ONE. After careful consideration, we feel that it has merit but does not fully meet PLOS ONE’s publication criteria as it currently stands. Therefore, we invite you to submit a revised version of the manuscript that addresses the points raised during the review process.

We look forward to receiving your revised manuscript.

Kind regards,

Mário Espada, PhD

Academic Editor

PLOS One

**Journal Requirements:**

1. When submitting your revision, we need you to address these additional requirements. Please ensure that your manuscript meets PLOS ONE's style requirements, including those for file naming. The PLOS ONE style templates can be found at https://journals.plos.org/plosone/s/file?id=wjVg/PLOSOne_formatting_sample_main_body.pdf and https://journals.plos.org/plosone/s/file?id=ba62/PLOSOne_formatting_sample_title_authors_affiliations.pdf 2. Thank you for stating the following financial disclosure: The author(s) declare that financial support was received for the research and/or publication of this article. We acknowledge the support by the KIT-Publication Fund of the Karlsruhe Institute of Technology.   Please state what role the funders took in the study.  If the funders had no role, please state: "The funders had no role in study design, data collection and analysis, decision to publish, or preparation of the manuscript." If this statement is not correct you must amend it as needed. Please include this amended Role of Funder statement in your cover letter; we will change the online submission form on your behalf. 3. We note that you have indicated that there are restrictions to data sharing for this study. For studies involving human research participant data or other sensitive data, we encourage authors to share de-identified or anonymized data. However, when data cannot be publicly shared for ethical reasons, we allow authors to make their data sets available upon request. For information on unacceptable data access restrictions, please see http://journals.plos.org/plosone/s/data-availability#loc-unacceptable-data-access-restrictions.  Before we proceed with your manuscript, please address the following prompts: a) If there are ethical or legal restrictions on sharing a de-identified data set, please explain them in detail (e.g., data contain potentially identifying or sensitive patient information, data are owned by a third-party organization, etc.) and who has imposed them (e.g., a Research Ethics Committee or Institutional Review Board, etc.). Please also provide contact information for a data access committee, ethics committee, or other institutional body to which data requests may be sent. b) If there are no restrictions, please upload the minimal anonymized data set necessary to replicate your study findings to a stable, public repository and provide us with the relevant URLs, DOIs, or accession numbers. Please see http://www.bmj.com/content/340/bmj.c181.long for guidelines on how to de-identify and prepare clinical data for publication. For a list of recommended repositories, please see https://journals.plos.org/plosone/s/recommended-repositories. You also have the option of uploading the data as Supporting Information files, but we would recommend depositing data directly to a data repository if possible. Please update your Data Availability statement in the submission form accordingly. 4. Please upload a new copy of Figure 1 as the detail is not clear. Please follow the link for more information:  https://journals.plos.org/plosone/s/figures 5. We note that this data set consists of interview transcripts. Can you please confirm that all participants gave consent for interview transcript to be published? If they DID provide consent for these transcripts to be published, please also confirm that the transcripts do not contain any potentially identifying information (or let us know if the participants consented to having their personal details published and made publicly available). We consider the following details to be identifying information:- Names, nicknames, and initials- Age more specific than round numbers- GPS coordinates, physical addresses, IP addresses, email addresses- Information in small sample sizes (e.g. 40 students from X class in X year at X university)- Specific dates (e.g. visit dates, interview dates)- ID numbers Or, if the participants DID NOT provide consent for these transcripts to be published:- Provide a de-identified version of the data or excerpts of interview responses- Provide information regarding how these transcripts can be accessed by researchers who meet the criteria for access to confidential data, including:a) the grounds for restrictionb) the name of the ethics committee, Institutional Review Board, or third-party organization that is imposing sharing restrictions on the datac) a non-author, institutional point of contact that is able to field data access queries, in the interest of maintaining long-term data accessibility.d) Any relevant data set names, URLs, DOIs, etc. that an independent researcher would need in order to request your minimal data set. For further information on sharing data that contains sensitive participant information, please see: https://journals.plos.org/plosone/s/data-availability#loc-human-research-participant-data-and-other-sensitive-data If there are ethical, legal, or third-party restrictions upon your dataset, you must provide all of the following details (https://journals.plos.org/plosone/s/data-availability#loc-acceptable-data-access-restrictions):a) A complete description of the datasetb) The nature of the restrictions upon the data (ethical, legal, or owned by a third party) and the reasoning behind themc) The full name of the body imposing the restrictions upon your dataset (ethics committee, institution, data access committee, etc)d) If the data are owned by a third party, confirmation of whether the authors received any special privileges in accessing the data that other researchers would not havee) Direct, non-author contact information (preferably email) for the body imposing the restrictions upon the data, to which data access requests can be sent 6. If the reviewer comments include a recommendation to cite specific previously published works, please review and evaluate these publications to determine whether they are relevant and should be cited. There is no requirement to cite these works unless the editor has indicated otherwise. 

**Additional Editor Comments:**

Dear Authors,

Congratulations on your work.

Please revise the manuscript in light of the reviewers' minor suggestions.

Thank you.

Best regards.

Reviewers' comments:

**Comments to the Author**

1. Is the manuscript technically sound, and do the data support the conclusions?

Reviewer #1: Yes

Reviewer #2: Yes

2. Has the statistical analysis been performed appropriately and rigorously?

Reviewer #1: Yes

Reviewer #2: N/A

3. Have the authors made all data underlying the findings in their manuscript fully available?

Reviewer #1: Yes

Reviewer #2: Yes

4. Is the manuscript presented in an intelligible fashion and written in standard English?

Reviewer #1: Yes

Reviewer #2: Yes

**Reviewer #1:**  Manuscript ID: PONE-D-25-53850

“Rethinking performance crises in professional 30 soccer: Coaches’ insights into systemic vulnerabilities and escalating dynamics”

A. Summary of the Research and Overall Impression

Thank you for the opportunity to review this paper. This manuscript presents a timely and conceptually rich qualitative exploration of how professional soccer coaches perceive the development and maintenance of performance crises. Anchored in an abductive, theory-informed thematic content analysis, the study advances current knowledge by integrating organizational, interpersonal, and psychological perspectives into a multi-level framework of crisis development. The explicit focus on the coaches’ standpoint, which has been comparatively underrepresented in the existing literature, constitutes a notable contribution.

Overall, the manuscript is theoretically well grounded, methodologically coherent, and empirically insightful. The integration of complexity theory, ecological systems theory, and sensemaking perspectives yields a nuanced understanding of how pre-crisis vulnerabilities, acute triggers, and escalating dynamics interact over time. At the same time, certain aspects of the presentation, particularly the density and layering of the findings section, the positioning of the study within the broader exploratory research tradition, and some minor issues in the reference formatting, would benefit from refinement. With revisions addressing these points, I believe the paper has strong potential to make a meaningful contribution to the literature on performance crises in professional sport.

Strengths

1. Provides an original and much‑needed focus on professional coaches’ perspectives, complementing player‑centred research on performance crises.

2. Develops a multi‑level, theoretically informed framework that links pre‑crisis vulnerabilities, crisis triggers, and escalating dynamics across organizational, team, and individual levels.

3. Offers practically relevant implications for organizational development, leadership alignment, and resilience-building in professional football.

Weakness

1. The Introduction could more explicitly integrate the key conceptual and theoretical frameworks (e.g., ecological systems, complexity in team dynamics) to create stronger continuity between the literature review, and the findings/discussion/conclusion.

2. The findings/results section is at times overly complex, with multiple nested layers and partially inconsistent terminology, which may challenge reader comprehensibility

3. The manuscript could position its exploratory qualitative design more explicitly within established methodological literature and clarify the use (or non-use) of formal intercoder agreement indices.

4. Minor but systematic issues in reference formatting (Vancouver style, capitalization, consistency across source types, internet references) require careful revision.

Recommendation

Overall, the manuscript makes a valuable contribution to the field and is suitable for publication after minor revisions.

B. Discussion of Specific Areas for Improvement

Major Issues

No major issues have been identified.

Minor Issues

• INTRODUCTION: In my view, a separate “Theoretical framework” section is not strictly necessary. However, it might be beneficial to enrich the latter part of the Introduction so that the main conceptual and theoretical underpinnings of the study are articulated more explicitly. For instance, the process-oriented models of performance crises, together with the ecological systems perspective, complexity theory, and sensemaking approach that are later used in the Discussion, could be more clearly introduced and integrated in the literature section. Such an expansion would not require adding an entirely new section, but rather refining the existing Introduction to ensure stronger conceptual continuity between the literature review, the findings, and the subsequent discussion. This would enhance the overall coherence of the manuscript and align it more closely with conventions in qualitative research reporting.

• The text convincingly establishes the aim and relevance of the study, particularly by demonstrating that the coach’s perspective has been largely neglected in previous research. A noteworthy point, however, appears in lines 110-114, where it is stated that clubs often enter cycles of failure without sufficiently learning from past crises and frequently resort to coach dismissals as the primary solution. I find this observation well grounded, yet I believe that briefly addressing the possible economic and sociological dimensions of this tendency could add further depth to the manuscript.

• For instance, designating the coaching staff as the “scapegoat” may represent a less costly and more “secure” option, both economically and symbolically, than confronting a squad of 22-30 players (lines 716-717 may perhaps offer a subtle hint here). Since fan identification and emotional attachment are largely formed through players, dismissing the coach rather than sanctioning or replacing allegedly problematic players may be perceived as a less controversial move in the eyes of supporters and the media. Moreover, when players are conceptualized as financial assets-linked to transfer fees, bonuses, and broadcasting-related revenues, clubs may consider it strategically more rational to replace the comparatively less protected coach than to “kill the goose that lays the golden eggs.” In addition to the financial burden of terminating contracts, asymmetric power relations may also arise from the potential legal disputes and further crises that could follow as stated at lines 636-638. (with the possible exception of very influential agents or highly established head coaches) I believe that briefly discussing these economic and sociological dynamics in the Discussion section could help to explain more comprehensively the decision logic underlying coach changes during performance crises. The treatment of this point in the Discussion section (lines 303–308) appears somewhat superficial. Expanding on this aspect with deeper conceptual or empirical insights would, in my view, substantially enhance the interpretive depth and overall clarity of the study.

• METHODS: The methods section is generally clear, systematic, and provides an adequate level of detail for a qualitative exploratory research design employing thematic content analysis. The integration of data-driven and concept-driven analytic approaches within an abductive (intuitive–deductive) logic offers a well-suited framework for exploring coaches’ perceptions of performance crises, consistent with the theoretical framework established by Jekauc et al. (2024).

• The application of the SRQR checklist, together with careful attention to trustworthiness criteria (Elo et al., 2014) and consensus coding procedures, demonstrates methodological rigor appropriate for exploratory qualitative designs. However, to further strengthen the methodological positioning, it would be beneficial for the authors to situate their abductive thematic content analysis more explicitly within the broader exploratory research tradition. For instance, brief references to foundational sources such as Lederman (1993) or Stebbins (2001) could help anchor the study’s methodological approach more firmly within the established exploratory research literature. While the reference to Rausch et al. (2025) provides valuable contextualization of the research group’s philosophical orientation (ontological relativism/constructivist epistemology), linking the research design, data collection strategy, and analytic procedures more explicitly to widely recognized methodological cornerstones in the qualitative literature would, in my view, further enhance perceptions of rigor and better situate the study within the broader qualitative research tradition.

• In addition, although the manuscript clearly describes the use of consensus coding, it remains unclear whether any formal index of intercoder agreement (e.g., Cohen’s kappa or a similar statistic) was calculated. Clarifying whether such an index was computed, and, if so, reporting its value, would further strengthen the transparency and trustworthiness of the analytic procedures.

• RESULTS: First of all, as this is a qualitative study, I would kindly suggest using the term “Findings” instead of “Results” for the corresponding section heading. In qualitative research, “findings” is more commonly used to emphasize the interpretive, meaning-oriented nature of the analysis, whereas “results” is more closely associated with quantitative, statistically driven outputs. This is, of course, only a recommendation, and I leave it to the authors’ discretion whether they consider this change appropriate for their manuscript.

• Figure 1 presents the overall model; however, the structure described in the text appears highly complex, with multiple nested layers. For example, under *Pre-crisis vulnerabilities*, there are *external vulnerabilities*, followed by subcategories such as *public expectations*, each of which is then elaborated with detailed explanations. I fully understand and appreciate your effort to present the richness of the findings. Considering that the journal’s publication format does not use numbered headings, the current structure contains too many nested layers and inconsistent naming conventions, making it challenging for readers to follow. Would it perhaps be sufficient to outline the general framework of the model and refer briefly to the main categories instead of providing such an extensive level of detail? I offer this as a suggestion, and I look forward to reading your reflections on this point in your response doc.

• I would like to highlight the finding presented in lines 491-506 as one of the most striking and insightful parts of the manuscript. Including a brief reference to this key result in the abstract could further strengthen the overall impact of the study and enhance its citation potential.

• DISCUSSION: In lines 763-770, the manuscript touches on what I consider one of its most original and compelling contributions. In some cases, attempts to resolve a problem are delegated to the very actors who contributed to its emergence, which can result in ineffective efforts and a vicious cycle of unproductive work. Within the football industry in particular, such unsuccessful interventions may not only cause psychological strain among fans but also lead to tangible and intangible losses for other stakeholders, including club employees at various levels, sponsors, and associated partners.

• It should be remembered that, regardless of whether football clubs are publicly supported or privately funded, they operate as legal entities in the contemporary sports industry with an influence that often exceeds their physical size or local base. Consequently, their failures can function like a “black hole,” drawing in and adversely affecting those around them. From this perspective, systematically identifying and diagnosing problems must be regarded as a crucial first step in any problem-solving process. All stakeholders, sometimes including the fans, need to demonstrate the willingness and capacity to assume responsibility when necessary. I would like to commend the authors for addressing these issues and for the considerable effort invested in this study.

• Strengths and limitations: In the “Strengths and limitations” section (line 855 onwards), the authors note that the exclusive focus on German football culture may limit the generalizability of the findings to other cultural contexts. However, it is well established that the primary aim of qualitative research is not statistical generalization, but rather the in-depth exploration and illumination of a given phenomenon within its specific context. From this perspective, the present study successfully uncovers a particular reality within the context of the German football tradition, but should not claim to explain structures and processes across the entire football industry. Instead of framing this as a limitation in terms of generalizability, it might be more consistent with the qualitative research design to emphasise how the findings and conclusions are interpreted in relation to the specific context under study. I would therefore kindly suggest that the authors explicitly foreground this contextual focus, either in the title and/or in the statement of the research aim in the introduction, so that the boundaries of the study are clearly delineated for the reader.

• REFERENCES: I would kindly recommend revising the reference list to ensure full compliance with the Vancouver style used by PLOS ONE. There appear to be some minor inconsistencies and deviations in the formatting of internet sources, the initial letters of journal names should be capitalized, and the same type of source (e.g., books) is presented in different formats and some more…

**Reviewer #2:**  I would like to thank the editors for the opportunity to review the manuscript entitled “Rethinking performance crises in professional soccer: Coaches’ insights into systemic vulnerabilities and escalating dynamics.” I also congratulate the authors for addressing a timely and relevant topic within the field of professional sport, offering an in-depth and thoughtful analysis of performance crises from the perspective of coaches.

Overall, the manuscript is well written and presents a clear and coherent structure. The rationale is well articulated, the research question is logically derived from the literature, and the methodological approach is appropriate and consistent with the study’s objectives. The alignment between the theoretical framework, data collection, and analysis strengthens the credibility and contribution of the work.

That said, I would like to raise just one point for clarification. It is not clear whether the study refers to professional men’s soccer, women’s soccer, or both. This ambiguity appears both in the title and in the methods section, where the gender of the sport context and of the participants (coaches) is not explicitly stated. This information is highly relevant, as the structural conditions, organizational contexts, and potential sources of performance crises may differ substantially between men’s and women’s professional soccer.

Clarifying whether the study focuses on men’s soccer, women’s soccer, or a mixed context—and specifying the gender of the participants—would significantly strengthen the manuscript, improve transparency, and enhance the interpretation and transferability of the findings.

**Do you want your identity to be public for this peer review?** For information about this choice, including consent withdrawal, please see our Privacy Policy

Reviewer #1: No

Reviewer #2: No

---

## [Author Response · Author response to Decision Letter 1]

2 Feb 2026

We sincerely thank the Academic Editor and Reviewers for their constructive and detailed feedback, which has significantly strengthened the manuscript. Please see the attached document "Response to Reviewers" for a detailed point-by-point reply to all reviewer comments.

---

## [Decision Letter · Decision Letter 1]

16 Feb 2026

Rethinking performance crises in professional soccer: German coaches’ insights into systemic vulnerabilities and escalating dynamics

PONE-D-25-63850R1

Dear Dr. Constantin Rausch,

We’re pleased to inform you that your manuscript has been judged scientifically suitable for publication and will be formally accepted for publication once it meets all outstanding technical requirements.

Kind regards,

Mário Espada, PhD

Academic Editor

PLOS One

---

## [Editor Report · Acceptance letter]

PONE-D-25-63850R1

PLOS One

Dear Dr. Rausch,

I'm pleased to inform you that your manuscript has been deemed suitable for publication in PLOS One. Congratulations! Your manuscript is now being handed over to our production team.

Kind regards,

on behalf of

Dr. Mário Espada

Academic Editor

PLOS One